Methods

# Evaluation of colorectal cancer subtypes and cell lines using deep learning

Jonathan Ronen[1,2] , Sikander Hayat[3], Altuna Akalin[1]

**Colorectal cancer (CRC) is a common cancer with a high mortality rate and a rising incidence rate in the developed world. Molecular profiling techniques have been used to better understand the variability between tumors and disease models such as cell lines. To maximize the translatability and clinical relevance of in vitro studies, the selection of optimal cancer models is imperative. We have developed a deep learning–based method to measure the similarity between CRC tumors and disease models such as cancer cell lines. Our method efficiently leverages multiomics data sets containing copy number alterations, gene expression, and point mutations and learns latent factors that describe data in lower dimensions. These latent factors represent the patterns that are clinically relevant and explain the variability of molecular profiles across tumors and cell lines. Using these, we propose refined CRC subtypes and provide best-matching cell lines to different subtypes. These findings are relevant to patient stratification and selection of cell lines for early-stage drug discovery pipelines, biomarker discovery, and target identification.**

## Introduction

Colorectal cancer (CRC) accounts for 10% of cancer-related deaths (1), with ~1.4 million new cases and 693,900 deaths reported in 2012 (2). CRC is not homogeneous and can be classified into different subtypes based on molecular and morphological alterations (3). The disease occurs when normal epithelial cells acquire genetic and epigenetic alterations that transform them to cancer cells. Mutations in the *WNT* signaling pathway are thought to initiate the transformation to cancer (4, 5). This is followed by deregulation of other signaling pathways such as *MAPK*, *TGF-β*, and *PI3K–AKT* via acquired mutations (5, 6). Since the original description of the molecular pathogenesis of CRC, multiple additional pathways, mutations, and epigenetic changes have been implicated in the formation of CRC (7). Based on integrative analysis of genomic aberrations observed in The Cancer Genome Atlas (TCGA) samples, a distinction can

be made between hypermutated (≈16%) and non-hypermutated (≈84%) CRC. The hypermutated cancers exhibit microsatellite instability (MSI), resulting from defective mismatch repair or DNA polymerase proofreading mutations (3). Non-hypermutated microsatellite stable (MSS) cancers are characterized by chromosomal instability (CIN), with high occurrence of DNA copy number alterations and mutations in the *APC*, *TP53*, *KRAS*, and *BRAF* genes (3). Most CRC tumors have aberrantly methylated genes, a subset of which may play functional roles in CRC (7). A further subset of CRC tumors display a CpG island methylator phenotype (CIMP), in which some tumor suppressor genes may be epigenetically inactivated (8). The diversity of molecular disease mechanisms results in distinct molecular subtypes of CRC, associated with different survival rates and responses to therapy. Hence, molecular subtypes can provide more clinically relevant information than standardised tumor staging based on the size and metastasis. A number of molecular subtyping schemes based on gene expression profiles were recently studied and summarized as the consensus molecular subtypes (CMS) (9), designating four main CRC subtypes with distinguishing features. The CMS1 subtype is defined by hypermutation, MSI, and strong immune activation; CMS2 is defined by CIN, WNT, and MYC signaling activation; CMS3 is defined by metabolic dysregulation; and CMS4 is defined by growth factor *β* activation, stromal invasion, and angiogenesis (9). This leaves ~13% of the tumors that cannot be assigned to a consensus subtype, as they have mixed gene expression signatures. These may represent distinct tumor subtypes or samples with intratumor heterogeneity. The CMS classification is based on gene expression, yet follow-up analysis of the tumors revealed distinct copy number profiles, mutation frequencies, and methylation profiles (9) (Fig S6), indicating that much can be gained from integrating other types of omics data.

We propose a multiomics method that incorporates data on gene expression, copy number, and mutations in identifying CRC subtypes, in a manner that has important implications for patient stratification. Our method is able to match cell lines to each subtype, and in the future, it will be useful in assigning best-matching xenografts or organoid models to the study of each subtype. Patient-derived tumor models have been shown to predict in patients' response to treatments (10), and so by finding best-matching tumor

[1]Max-Delbrück-Centrum für Molekulare Medizin, BIMSB, Berlin, Germany  [2] Humboldt Universität zu Berlin, Berlin, Germany  [3]Bayer AG, Department of Bioinformatics, Berlin, Germany

Correspondence: altuna.akalin@mdc-berlin.de

models, we hope to empower their clinical use without the need to grow new organoids from each patient. Using multiomics data sets permits subtyping of CRC tumors that have not been associated with a CMS subtype, which is based on gene expression only. The multiomics signatures, incorporating gene expression, point mutations, and copy number alterations, are a direct output of the method and do not need to be generated post hoc, for example, by examining mutation rates in groups defined by gene expression profiles, as is the case with CMS. With these goals in mind, we used deep learning on multiomics data to refine CRC subtypes in an unsupervised manner and then match them to cell lines.

Genomic assays are high dimensional (tens of thousands of genes), and high-dimensional spaces are challenging to analyze. This problem is further exacerbated when seeking to integrate multiple types of assays from diverse omics platforms. It is, therefore, beneficial to introduce methods which are able to recognize patterns in the molecular signatures spanning different omics data types. Latent factor analysis is an unsupervised learning technique well suited to the task at hand. This type of analysis is sometimes referred to as dimensionality reduction; it seeks to learn (infer) a lower dimension representation of data, which preserves the important structure/pattern therein. The method describes data using a handful of latent factors, rather than tens of thousands of genes, simplifying downstream analysis such as distance calculations and clustering (11). In the context of multiomics data analysis, latent factors may be thought of as high-order genomic features. Instead of examining point mutations or expression values of single genes, latent factors summarize patterns that span different data types.

The patterns represented by latent factors should be interpretable in the biological context, that is, it is desirable that the patterns correspond to cellular processes. Matrix factorization has become the workhorse for latent factor analysis for multiomics and general data analysis (12). Latent factor analysis for multiomics data typically includes concatenating different omics data to a single matrix and applying a well-known matrix factorization algorithm, sometimes with weighting of the individual data sets. Multifactor analysis (MFA) (13) and iCluster+ (14) are examples of such methods. Some such algorithms, such as MFA, impose orthogonality of factors, that is, the factors explain disjoint underlying processes, as in PCA. Orthogonality might be conceptually appealing, but is not a biological necessity. Orthogonal latent factors may be the best for statistical reconstruction of a data set and still be biologically difficult to interpret. On the other end of the spectrum, deep learning–based methods work as dimensionality reduction techniques that can deal with nonlinearity and can generalize well on a number of problems. In addition, they can be sparse: each latent factor depends only on a few of the input genes, and each tumor is described by only a handful of latent factors. This sparsity simplifies the task of biological interpretation of the model by pointing out specific biological processes underlying the latent factors; conversely, sparsity in the relationship between latent factors and tumors simplifies downstream analyses such as clustering.

Although it is reasonable to expect that the latent factors describing the data be of much lower dimensionality than the

genome-scale input data, it is an open question just how low the dimensionality should be. Heuristics to pick the number of latent factors have been proposed by method designers. PCA and MFA typically suggest an elbow method, where latent factors are ordered by their variance explained, and the user determines an elbow point in the graph, discarding latent factors with a low variance explained. MOFA formalizes this heuristic by starting the fitting process with a high number of latent factors, and during training, discarding ones with a variance explained below a preset threshold (with a default value of 2%). iCluster+'s heuristic comes from its k-means roots, that is, one tends to set the number of latent factors to $K - 1$, where $K$ is the number of clusters one expects to find. Recently, a large-scale study of latent factor methods (15 *Preprint*) has demonstrated the desirability of specifying more latent factors than are expected to exist in the data. Specifically, it was shown that using more latent factors than are needed can compensate for shortcomings in the training algorithm and improve log-likelihood and recovery of ground truth latent factors. Hence, it is desirable to have a latent factor method that is able to learn a large number of latent factors efficiently from data.

With these requirements in mind, we used a variational autoencoder (VAE) (16 *Preprint*), a flexible framework for nonlinear latent factor inference. Our implementation is inspired by methods implemented by ladder-VAE (17 *Preprint*) and disentangled autoencoders (β-VAE) (18). Using these adaptations over a "vanilla" VAE enables the model to converge quickly to a good representation, in spite of the seemingly large complexity of the problem space. Such deep learning frameworks are flexible and have been used to deal with many kinds of data sets (19 *Preprint*). A similar method proved successful at stratifying cancers by their tissue type based on gene expression profiles (20 *Preprint*), and other autoencoder architectures have been used to integrate multimodal data in robotics (21), as well as protein function prediction (22).

Here, we used a multimodal, stacked VAE to extract latent factors that can be used to identify CRC subtypes and predict patient survival. We call the method, "multiomics autoencoder integration," or maui. We compared maui performance with state-of-the-art multiomics analysis methods and showed that it outperforms the matrix factorization methods, showing improved clinical relevance. Computationally, maui's is in orders of magnitude more efficient than the state-of-the-art methods we compared it with. In a further step, we map commonly used CRC cell lines to the latent factor space defined by maui. This allows us to use patterns recognized by maui as significant in CRC to rank the suitability of in vitro models for the study of specific tumors. We, then, determined the cancer cell lines most appropriate for drug target studies aimed at distinct CRC subtypes.

## Results

### Refining CRC subtypes using multiomics data

The CRC cohort in the TCGA data set (n = 519, see the Materials and Methods section) has been extensively studied. Statistical analysis

**Table 1. Summary of TCGA tumors' CMS labels.**

| CMS label | Description | No. of samples |
|---|---|---|
| CMS1 | MSI immune | 61 |
| CMS2 | Canonical | 175 |
| CMS3 | Metabolic | 60 |
| CMS4 | Mesenchymal | 123 |
| Total with CMS label | | **419** |
| Without CMS label | | 100 |
| Total | | **519** |

Bold entries indicate sums.

supports the use of clinical follow-up data (23), and a state-of-the-art subtyping scheme is described in the "consensus molecular subtypes" for CRC or CMS (9) (Table 1). To validate the relevance of latent factors learned by maui to cancer biology, we tested the extent to which these latent factors recapitulate the known subtypes.

We used maui to extract latent factors from data on gene expression, point mutations, and copy number alterations and did the same using MOFA (24) and iCluster+ (14), other published methods for multiomics integration by dimensionality reduction. We also used PCA as a baseline for integrating multiomics data (see the Materials and Methods section). We based our comparison of the methods on a quantification of the relationship between latent factor representations and the CMS subtype, using support vector machines (SVMs) to predict the CMS subtype from the latent factor representation. We used regularized linear SVM, where the regularization parameter was selected using cross-validation (CV). Also using CV, we predicted the CMS label of each tumor out-of-sample (when it was in the validation fold, see the Materials and Methods section). We then computed "receiver operating characteristics" (ROCs) and the area under the curve (see the Materials and Methods section). The area under the ROC (auROC) is a measure of classification accuracy. A random guess is expected to deliver a score of 0.5, and 1.0 would represent perfect prediction. All three methods produce latent factors with predictive power of the CMS labels above random guessing (Figs 1A and S1A). Using the SVM, maui (auROC 0.98) marginally outperforms MOFA (auROC 0.94), and both dramatically outperform iCluster+ (auROC 0.73) and PCA (auRoc 0.85) (Figs 1B, S1B, and S2).

In this analysis, maui had an advantage over MOFA, in that it was run with 80 latent factors, whereas MOFA only generated 20 (MOFA uses heuristics to pick the number of latent factors). Supervised learning algorithms such as SVM may benefit from a larger number of input features (here, the latent factors). We would like to demonstrate that maui's ability to learn more latent factors (owing to its superior computational efficiency) is beneficial in a fair benchmark. To assess which of the methods is best able to capture the CMS labels, regardless of the number of latent factors, we repeated the previous exercise—predicting the CMS from the latent factors—using an unsupervised learning algorithm. We clustered the samples with a well-defined CMS (some CRC samples do not have a CMS) using k-means clustering on the latent factors (see the Materials and Methods section). We let $K$ to vary

from 2 to 9, and for each $K$, we computed adjusted mutual information (AMI) of the clustering with the CMS labels. k-means clustering only reproduces the CMS subtype to a significant degree for $K$ values of 4–6 only using maui (Figs 1C and S1C). This analysis shows that maui factors are superior at predicting CMS labels, in a fair comparison, as k-means clustering is based on distances, whose computation does not benefit from higher dimensionality—in fact, the opposite is true (11).

Latent factors inferred by maui are predictive of the CMS subtype using k-means, especially using $K$'s 4–6 (Fig 1C). To pick the best clustering result to focus on, we computed the log-rank statistic for the significance of the differential survival rates between clusters (see the Materials and Methods section). $K = 6$ results in the most statistically significant survival difference ($P < 0.001$, Figs 1D and S3C). Note that the CMS subtypes on their own are not indicative of survival rates in the TCGA data ($P = 0.77$ Fig S3A) and that $K = 4$ ($P < 0.045$ Fig S3D) and $K = 5$ ($P < 0.019$ Fig S3B) also produce clusters with significant differential survival rates. Notably, $K = 6$ is preferable to $K = 4$ and $K = 5$, as it is able to tease out a cluster of patients with particularly poor prognoses (cluster 3); this consists mainly of a subset of tumors designated as CMS2 (canonical) (Fig S3). K-means clustering of MOFA or iCluster+ latent factors do not produce statistically significantly separable clusters (Figs S4 and S5).

We also compared the ability of maui, MOFA, and iCluster+ to predict patient survival, irrespective of any clustering. For each model, we first selected a subset of latent factors which are individually predictive of patient survival, calling those clinically relevant latent factors. This was carried out by fitting univariate Cox proportional hazards regression models, one per latent factor, and selecting ones for which the coefficient is nonzero with $P < 0.05$ (see the Materials and Methods section). Using those clinically relevant latent factors, we fitted a multivariate Cox proportional hazards regression and computed Harrell's c-index (25) (see the Materials and Methods section). The c-index is a measure of prediction accuracy for censored data, with a score of 0.5 expected for random guessing and 1.0 representing perfect accuracy. maui ($c = 0.72$) outperforms MOFA ($c = 0.68$), iCluster+ ($c = 0.64$), and PCA ($c = 0.64$) in this benchmark (Figs 1E and S1D).

The CMS subtyping scheme, as well as much of the work in the field, is based solely on gene expression profiles. To examine whether maui gives better predictions of patient survival with the addition of mutations and copy number data, we also trained a maui model based on gene expression alone. The maui model based on expression alone ($c = 0.69$) achieves a lower score than a maui model with multiomics data ($c = 0.72$), even when the former is provided with more genes as input features (Fig 1F). This indicates that data other than transcriptomes do contribute to overall performance of maui.

One advantage of maui over other methods such as iCluster+ and MOFA is that it is able to learn orders of magnitude for more latent factors, at a fraction of the computation time (Table 2). To demonstrate the advantage of being able to fit larger models, we also trained a maui model based on 6,000 multiomics features (see the Materials and Methods section). That model ($c = 0.75$) outperforms the smaller model (Fig 1F), demonstrating the

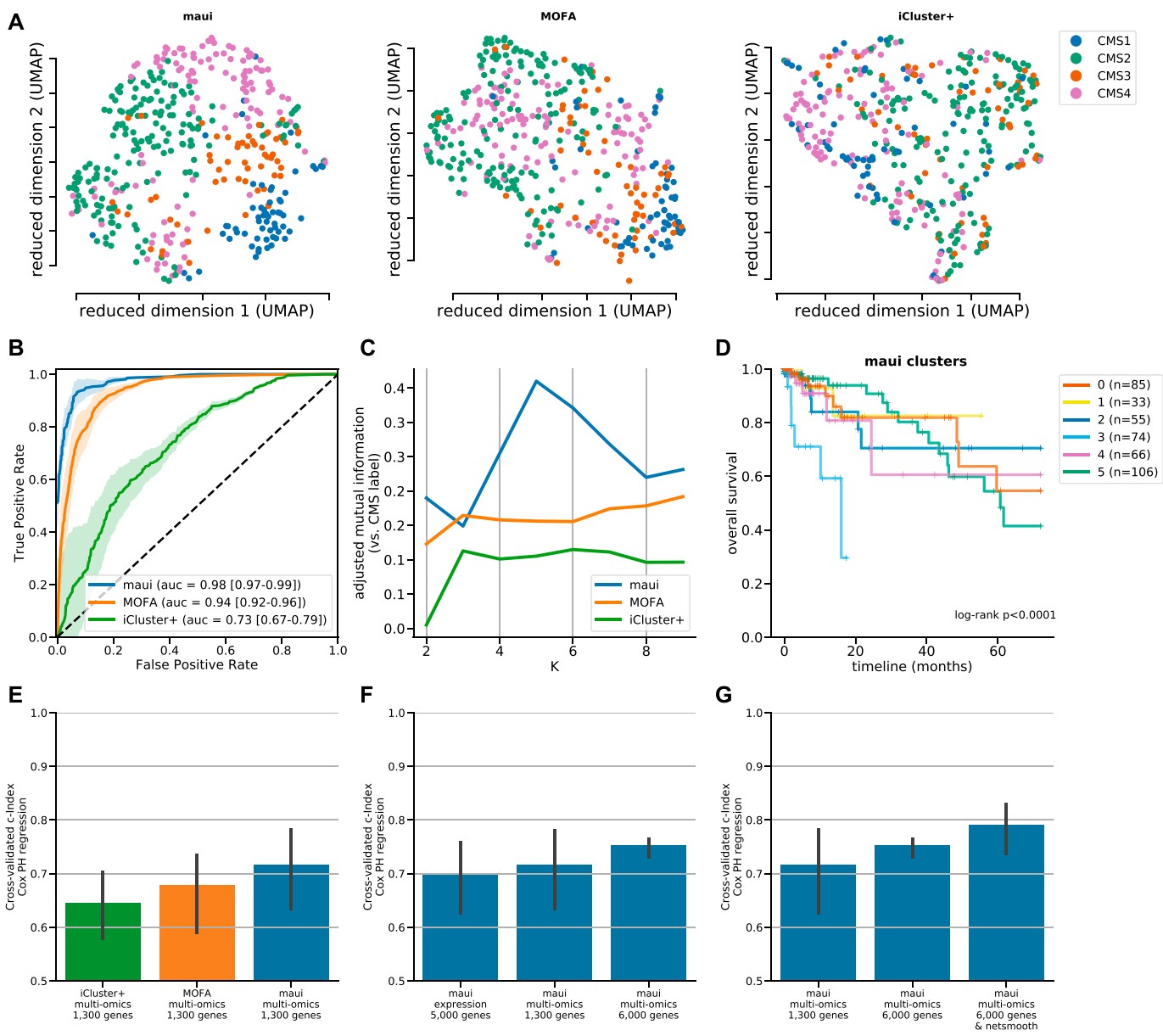

**Figure 1. maui, MOFA, iCluster+, and the CMS labels.**
**(A)** UMAP (29) reduced dimensions from latent factors inferred by maui, MOFA, and iCluster+. Each dot represents a tumor, colored by their CMS label. **(B)** ROCs for regularized SVMs predicting the CMS label from latent factors (out-of-sample, 10-fold CV). Mean ROC shown (see the Materials and Methods section) (C) The AMI (see the Materials and Methods section) of clusters obtained from latent factors inferred by maui, MOFA, and iCluster+, using k-means clustering with K ranging from 2 to 9. **(D)** Kaplan–Meier estimates and the log-rank statistic for differential survival of different clusters. The reported P value is from a multivariate log-rank test, under the null hypothesis that all groups have the same survival function. Clusters 3 and 5 represent a novel splitting of a previously defined subtype, CMS2. **(E)** Harrell's c-index for Cox regressions of iCluster+, MOFA, and maui shows maui is more predictive of patient survival than other methods. **(F)** Harrell's c-index comparing different maui flavors shows that maui benefits from multiomics data, as well as from more input genes. **(G)** Harrell's c-index shows network smoothing of mutations improves survival prediction using maui.

clinical utility of learning from more input features and underlining the importance of the computational efficiency offered by maui.

Finally, we investigated the usefulness for CRC subtyping of using prior information from protein interaction networks. Several groups, including our own, have previously incorporated gene–gene interactions using a method called network smoothing (26, 27). This is accomplished by allowing binary mutation values to diffuse over a gene network, a process which assigns nonzero "mutation scores" rather than binary mutation values, to genes which either have mutations or interact with mutated genes. To carry this out, we used a gene network defining interactions between genes from the STRING db (28), a database of protein–protein interactions. We applied the *netSmooth* (27) algorithm (see the Materials and

**Table 2. Summary of methods.**

| Method | Notes | No. of factors | Runtime |
|--------|-------|---------------|---------|
| iCluster+ | Bayesian, MCMC | 10 | ~11 h |
| MOFA | Bayesian, variational | 20 | 20 min |
| maui | Multilevel Bayesian, Stochastic Gradient Descent | 100 | 3 min |

Reported runtimes are on a single core of an Intel(R) Core(TM) i7-7500U CPU @ 2.70 GHz, on an HP Spectre x360 laptop with 16 GB of ram. No graphics processing units were used to fit the neural network.

Methods section) to the mutation data before passing it to maui and computed Harrell's c-index, as above. This revealed that network smoothing mutations further improve the clinical relevance of latent factors learned when integrating multiomics data (c = 0.79) (Fig 1G). Other network sources than STRING db, or protein–protein interactomes (PPIs) in general, also result in improvements using *netSmooth*, whereas networks with similar characteristics but without real information about interactions do not (27).

A closer examination reveals the degree to which maui clusters resemble the CMS subtypes and where they diverge. CMS1 is captured by cluster 2, CMS2 is split between clusters 3 and 5, CMS3 is captured by cluster 0, CMS4 overlaps with cluster 4, and cluster 1 is mixed (Fig 2A–C). A similar conclusion can be reached based on a set of molecular indicators introduced in (9): CMS1 and cluster 2 show the hypermutated (Fig S6A), CIMP (Fig S6B), and microsatellite unstable phenotypes (Fig S6C). They also exhibit similar mutation rates among *TP53*, *APC*, *KRAS*, and *BRAF* (Fig S6D), a set of genes commonly mutated in CRCs.

This might explain a seeming contradiction in the results—maui clusters, which are predictive of patient survival (Fig 1D), largely recapitulate the CMS, which is not predictive of survival in this cohort (Fig S3A). This is possible because although there is much agreement between maui and the CMS, there are small discrepancies in each cluster (Fig 2C). The CMS subtypes were defined based on gene expression signatures alone, using a larger cohort of patients which also includes the TCGA cohort, used here. The larger cohort, however, does not have multiomics data, and so we were unable to test this result in the larger cohort (in which the CMS is predictive of survival).

Fig 2C and S6 beg the question of why CMS2 was split into two clusters (3 and 5). To investigate the biological plausibility of this distinction, we performed a differential expression analysis and identified marker genes for each cluster. We, then, ran these lists through a gene set enrichment analysis (see the Materials and Methods section). Each maui cluster turns out to be associated with a distinct set of pathways (Fig 2D). Specifically, cluster 3 is dominated by TGF-$\beta$ signaling and leukocyte migration, whereas cluster 5 is dysregulated in ErbB, Hippo, and Wnt signaling pathways. These findings demonstrate that groups are distinct and exhibit different molecular phenotypes. Further evidence comes from the fact that prognoses for patients in cluster 3 are worse than cluster 5 (log-rank $P$ < 0.001, Fig S7). Cluster 4 (CMS4) is enriched in pathways associated with mobility and structural differences (Fig 2D), which is consistent with the higher stromal infiltration found in patients with CMS4 (9).

## CRC latent factors are associated with processes related to tumor progression and development

The superior computational efficiency of maui enables it to infer a large number of latent factors from multiomics data. This provides an opportunity to select those that are most interesting and might serve as biomarkers. To demonstrate this, we fitted Cox proportional hazards models (29), fitting one regression model for each factor, as above, selecting clinically relevant latent factors (see the Materials and Methods section). Fig 3B shows the 95% confidence interval of coefficients for these latent factors, showing that high values for some of these latent factors are predictive of a poor prognosis ($\beta$ > 0), whereas others are predictive of more favorable outcomes ($\beta$ < 0). This lends a significant prognostic value to such latent factors.

Another use derives from an interpretation of what these biomarkers represent. maui is very powerful because it can learn highly nonlinear patterns. This comes at a certain cost: it makes the biological interpretation of factors less straightforward than in a linear matrix factorization approach, such as PCA or MOFA. These other methods learn linear relationships between genes and latent factors, of the form $x = Wz$, where $W$ is directly available. From $W$, the connections between latent factors and genes may be directly read. maui does not produce a straightforward, linear $W$, so to associate latent factors with input genes, we correlated input genes with latent factor values (see the Materials and Methods section). While most latent factors are active in the gene expression domain, mutation data do not significantly affect most of them. Some latent factors capture interactions between two or more omics types (Fig 3A). This correlation between latent factors and input features permits us to overcome the difficulties presented by the nonlinear relationships and use the associations to find biologically relevant interpretations for the neural latent factors.

When we associated clinically relevant (see the Materials and Methods section) latent factors with gene ids, we observed an enrichment of pathways such as Wnt signaling and other APC-mediated processes known to play a role in CRC (Fig 3C). One of the factors most significantly associated with survival is enriched in neuronal growth factor (NGF) signaling. NGF signaling, which controls neurogenesis, has been associated with aggressive colorectal tumors (30, 31). Other latent factors relevant to survival implicate PDGF signaling, which is associated with stromal invasion and poor prognosis for CRC patients (32, 33). Thus, in addition to exposing latent factor biomarkers with the prognostic value, maui sheds light on underlying biological processes that merit study in search for new drug targets.

## maui performance is robust to parameter choice

To check the robustness of maui and ensure we were not overfitting, we ran maui and iCluster+ with a range of parameters, such as the number of latent factors. MOFA uses a heuristic to pick the number of latent factors, starting with a large number, and during training, removing latent factors which explain the variance in the data below a threshold of 2%.

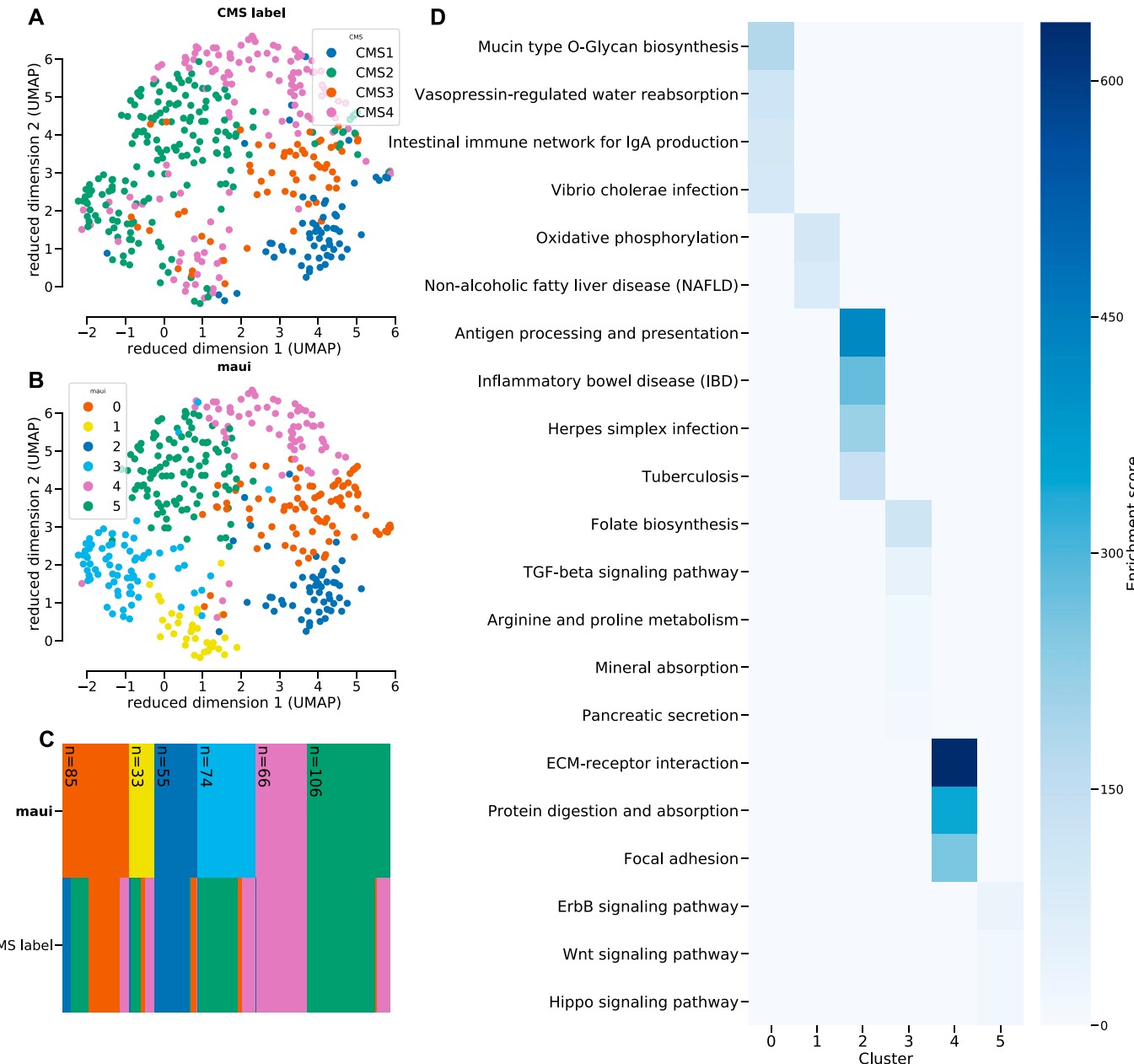

**Figure 2.   Clustering the tumors using k-means using the latent factors from maui reproduces the CMS labels closely, with the exception of CMS2 being split into two clusters, 3 and 5.**
**(A, B, C)** UMAP embedding of tumors colored by the CMS label, (B) UMAP embedding colored by k-means clusters on maui latent factors. (C) Cluster diagram depicts the correspondence between maui clusters and the CMS subtypes: the two rows represent the different labeling schemes (maui clusters and CMS subtypes), and each column represents a sample, which is colored according to its assignment in each row. The legend in subfigures (A, B) applies to the color scheme in (C) as well. **(D)** Pathways that are enriched in differentially expressed genes for each maui cluster. Clusters show a disjoint set of dysregulated pathways, underlining the different molecular phenotypes which underlie each group. Cluster 3 and 5 (which together make up the bulk of CMS2) are dominated by dysregulation of TGF-$\beta$ signaling and ErbB/Wnt/Hippo signaling, respectively.

We designed a compound benchmark score to test the overall performance of the model (see the Model selection section). We computed the compound benchmark score for each model using the different hyperparameter sets (see the Materials and Methods section). maui robustly outperforms iCluster+ ($P < 0.001$, one-sided $t$ test) and MOFA ($P < 0.01$, one-sided $t$ test) (Fig S11).

## Quality assessment of CRC cell lines as models for tumors

The molecular profiles of cancer cell lines often differ significantly from those of tumors because of the differences in selective pressures faced by cells in culture and natural tumor microenvironments; adaptation requires distinct genomic alterations (34). This means that not all colorectal-derived cancer

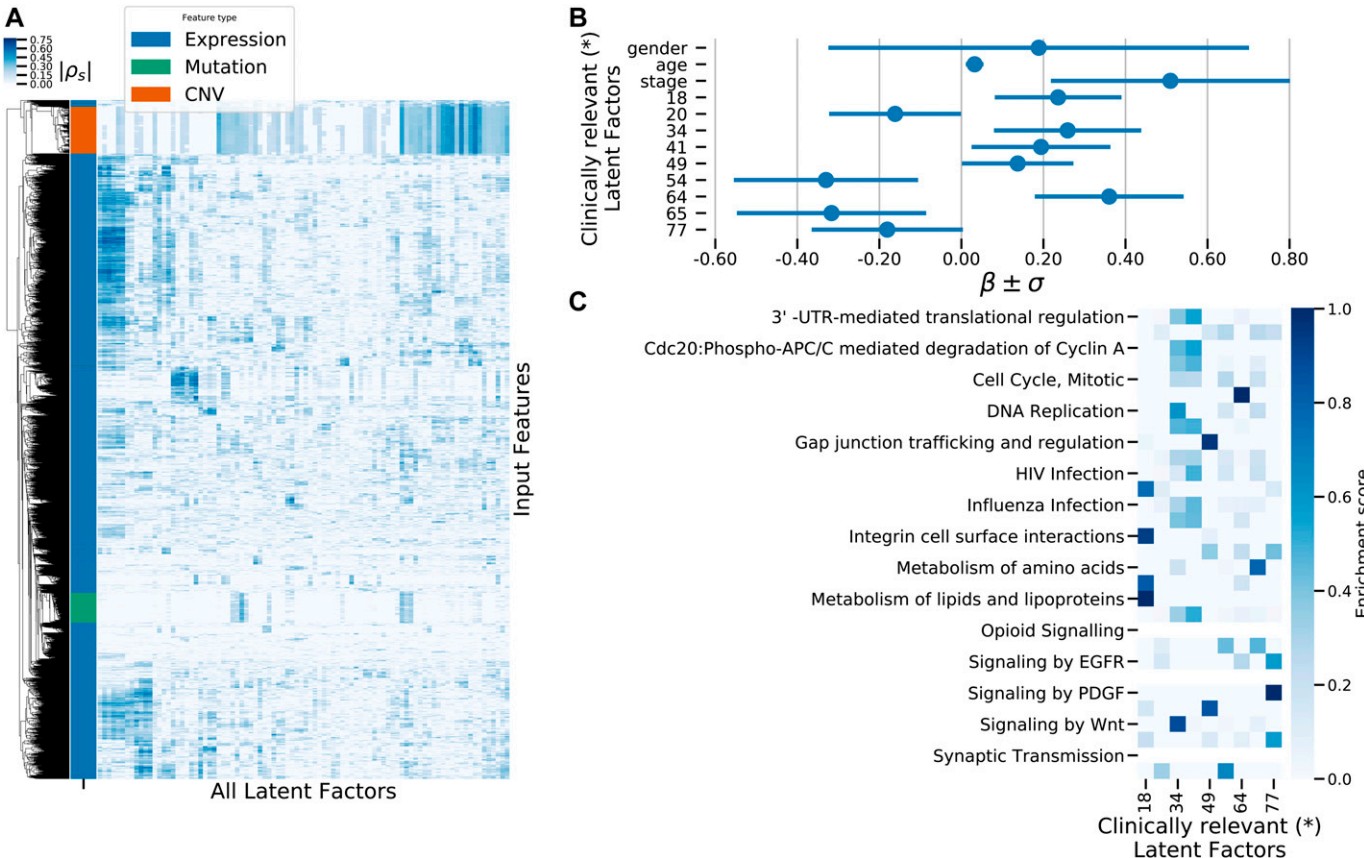

**Figure 3. Interpretation of maui latent factors.**
**(A)** A heatmap depicting the absolute correlation coefficients of the different input genes with the latent factors. Only input features with significant correlations ($P_{adj}$ < 0.01, see the Materials and Methods section) are depicted in the heatmap. The row annotation shows the type of input feature, that is, expression value, mutation, or copy number. **(B)** The coefficients in a Cox proportional hazards regression for factors which are clinically relevant (*) when controlling for patient age, sex, and tumor stage. Coefficients also shown for those covariates. **(C)** Pathway enrichment scores for genes associated with the latent factors which carry the prognostic value (have significant effects in Cox regression). (*) Clinically relevant factors are factors with a coefficient in a fitted Cox model controlling for age, sex, and tumor stage, which are statistically significantly nonzero ($P_{adj}$ < 0.05).

cell lines are likely to have equal value as models for tumors. Furthermore, over time, cancer cell lines run the risk of contamination and mislabeling. For instance, a cell line which was originally annotated as colorectal has been shown to be derived from melanoma (35). Because the identification and quality control of the cell lines are crucial steps in the research process, it is essential to know if the lines have diverged too much from tumors in their molecular makeup, been mislabeled, or contaminated. We examined 54 cancer cell lines derived from tumors of the colon from the Cancer Cell Line Encyclopedia (CCLE). We used maui to infer latent factor values for the cell lines to permit their characterization using the same latent factors as the tumors. As cell lines may develop adaptations specific to cell culture, their molecular profiles are often more similar to other cell lines than to those of primary tumors. We, therefore, hypothesized that cancer cell lines that are more similar to other cell lines than to tumors are less likely to be appropriate models for CRC tumors. We compiled a list of nearest neighbors (see the Materials and Methods section) for each cell line and, then, counted how many of its nearest neighbors are cell lines (as opposed to tumors). We used Euclidean distance in the space defined by the latent factors to determine similarity and found that

about half of the CRC cell lines we investigated belong to a "cell line cluster," meaning that most of their neighbors were other cell lines (Fig 4A). We eliminated cell lines where this proportion is above half and found among them a mislabeled cell line: *COLO741*, which has been shown to derive from melanoma and not from CRC (in more recent versions of the CCLE annotations, this has been fixed). This finding indicates the merit of using this method to flag cell lines as poor models for tumors.

In lieu of knowledge of other mislabeled or otherwise inappropriate colon-derived cancer cell lines, we artificially contaminated the data set by adding a random sample of 60 noncolon cell lines, assuming that these would be ill suited to the study of CRC tumors. The identities of these "known contaminant" cell lines are irrelevant, as we show later that the method works on 100 such random draws. We used this to repeat the exercise of counting the nearest neighboring cell lines. With the introduction of these true positives (non-colon cancer cell lines are considered true positives in the task of predicting which cell lines are poor models for CRC tumors), we found that more of the cell lines could be assigned to a "cell line cluster" in which most of their neighbors are other cell lines (Fig 4B). For nearly all noncolon derived cell lines, the

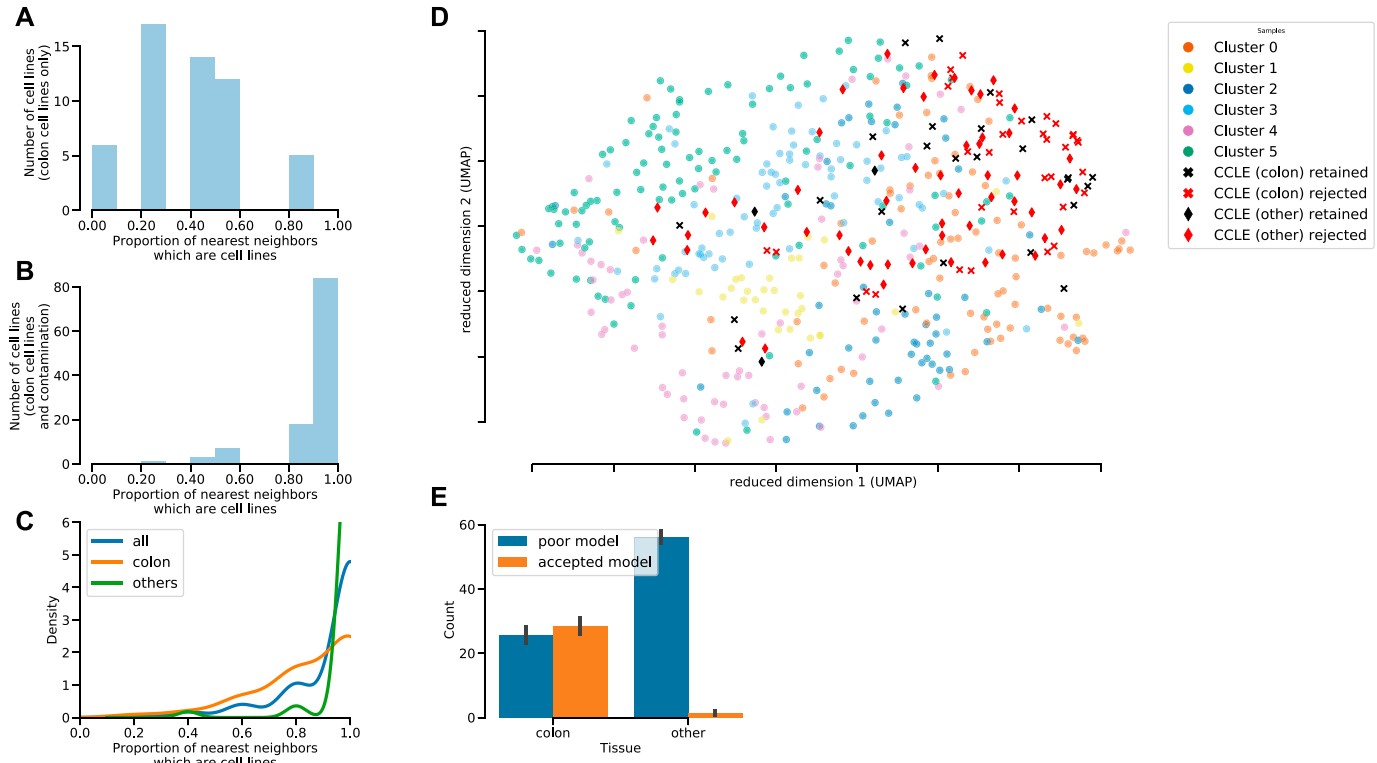

**Figure 4. For each cell line, we compiled a list of five nearest neighbors in latent factor space and counted the nearest neighbors that are cell lines (as opposed to tumors).**

Cell lines whose five nearest neighbors are all other cell lines are marked as less likely to be appropriate models for tumors, as they are more similar to cell lines than to tumors. Cell lines that had at least one nearest neighbor that is a primary tumor (not a cell line) were marked as likely good models. The nearest neighbors were calculated using the Euclidean distance in the space defined by the latent factors. Subfigure (D) above (UMAP projection of the latent factor space) does not always reflect the nearest neighbors, as UMAP gives no such guarantees. **(A)** Histogram of the proportion of nearest neighbors of cell lines which are also cell lines, CRCs only. **(B)** Histogram of the proportion of nearest neighbors of cell lines which are also cell lines, CRCs, and noncolorectal cell lines **(C)** Kernel Density Estimates of the proportion of cell-line neighbors among all cell lines (colorectal and noncolorectal), broken down by tissue. **(D)** UMAP embedding of tumors and cell lines. Crosses are colon-derived cell lines; diamonds are artificial contamination (noncolon-derived cancer cell lines). Red cell lines are rejected, and black ones are retained as more likely to be good models. **(E)** The proportions of colon and noncolon cell lines which are rejected because their proportion of nearest neighbor cell lines is above the threshold. Nearly all noncolon cell lines are consistently rejected, as well as about half of the colon cell lines.

five nearest neighbors were other cell lines, whereas this was not the case for colon-derived cell lines (Fig 4C). As a result, we designated cell lines whose five nearest neighbors are other cell lines, as less suitable for the study of colorectal tumors ("rejected"), as they more closely resemble other cell lines, even those derived from other tissues. We retain cell lines with at least one tumor among their five nearest neighbors as more likely to be suitable models. The choice of K = 5 for the number of nearest neighbors is immaterial, as the method is insensitive to the choice of K (Fig S8). UMAP embedding of the latent factor space of tumors (with CMS labels, n = 419), CRC cell lines (n = 54), and noncolorectal (artificial contamination, n = 60) cancer cell lines shows that this procedure eliminates most contamination cell lines, as well as some of the colon cancer cell lines, and that nonrejected cell lines are spread among all clusters (Fig 4D). Although, we advise caution when interpreting Fig 4D. The distances in 2D plot are not fully representative of the Euclidian distances and can be misleading as some cell lines look like rejected but they are near primary tumors in 2D. However, these cell lines are not near primary tumors when higher dimensions considered. We repeated the analysis with 100 more

random draws of 60 additional contaminants. For each such draw, we rejected any cell line whose five nearest neighbors are cell lines. This method consistently rejects almost all known contaminants and about half of the CRC cell lines (Fig 4E). Rejecting these cell lines is not necessarily a mistake because even if they originate in colon cancer, this does not guarantee they will be good genomic models for such tumors because of, for example, genomic divergence, mislabeling, or contamination. In addition, the fact that a particular cell line more closely resembles noncolon-derived cancer cell lines than CRC tumors is an indication that it might not be suitable as a model for CRCs. That this method successfully rejects almost all known contaminants is another indication that rejected colon cancer cell lines are likely to be poor models for CRC as well. The CRC cell lines CL40, SW1417, and CW2 are deemed most suitable as models for CRC tumors (Fig 5). Using the same criteria, the cell line COLO320 ranked among the lowest. COLO320 lacks mutations in major CRC driver genes such as *BRAF*, *KRAS*, *PIK3CA*, and *PTEN*, and it is actually of a neuroendocrine origin (36, 37). This very likely makes COLO320 a poor model for CRC.

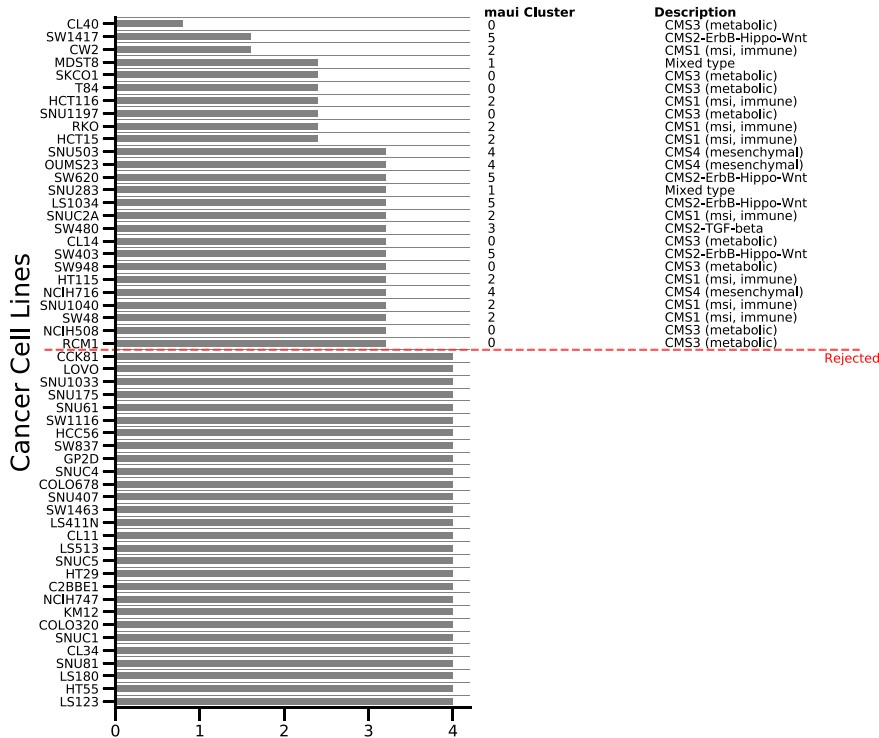

**Figure 5. For all colon-derived cell lines, we compiled lists of their five nearest neighbors.**
The barplot shows how many of those five were other cell lines. Cell lines where all five nearest neighbors are other cell lines are rejected, those having at least one nearest neighbor that is a tumor are kept and assigned to clusters, as shown in the table on the right.

## A complete subtyping scheme for CRC and appropriate cell lines for the study of each subtype

The CMS scheme (9) is incomplete because it is unable to classify many tumors. We used maui to assign the remaining non-CMS tumors to subtypes by repeating the clustering analysis while including both tumors that do not have a CMS designation and cancer cell lines. In this process, we included the cancer cell lines deemed to be suitable models (see above) to assign the cell lines to CRC subtypes. Here, we present a novel subtyping scheme for CRC, which covers the whole TCGA cohort and includes tumors without a CMS designation.

We also associate CRC cell lines with these subtypes. The tumors without a CMS label are distributed roughly according to the cluster size, as is to be expected for samples that lack a consensus definition (Fig 6A and B), and each cluster is associated with at least one cell line (Fig 6C and Table 3). The correspondence between our clusters and the CMS subtypes remains strong also when including cell lines and non-CMS tumors (Fig 6D). Cluster 2 (CMS1, MSI) is associated with the most cell lines; it comprises hypermutated tumors with low CIN. The cell lines that matched to cluster 2 show the same characteristics (Fig S9), another indication that latent factors capture patterns which are important to cancer biology. We believe that both the new

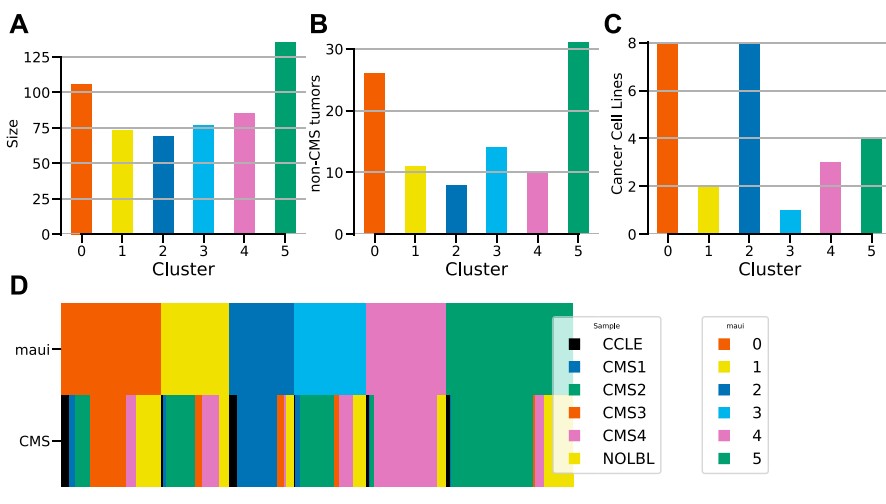

**Figure 6. Assignment of non-CMS tumors and cancer cell lines to clusters. (A)** The sizes (number of samples) of the clusters. **(B)** The number of non-CMS tumors assigned to each cluster. **(C)** The number of cancer cell lines associated with each cluster. **(D)** Cluster diagram depicts the correspondence between maui clusters and the CMS subtypes: the two rows represent the different labeling schemes (maui clusters and CMS subtypes), and each column represents a sample, which is colored according to its assignment in each row. The NOLBL samples without a defined CMS subtype are distributed among all clusters, as are cancer cell lines CCLE.

**Table 3.  maui clusters and the cancer cell lines associated with them.**

| Cluster | Description | Cell lines |
|---|---|---|
| 0 | CMS3 (metabolic) | SW948, CL14, SNU1197, RCM1, NCIH508, CL40, T84, SKCO1 |
| 1 | Mixed type | SNU283, MDST8 |
| 2 | CMS1 (msi, immune) | CW2, HT115, SNU1040, HCT15, SW48, HCT116, RKO, SNUC2A |
| 3 | CMS2-TGF-beta | SW480 |
| 4 | CMS4 (mesenchymal) | OUMS23, SNU503, NCIH716 |
| 5 | CMS2-ErbB-Hippo-Wnt | SW403, LS1034, SW620, SW1417 |

classification system and the assignment of best-fitting model cell lines will be a useful resource for future drug discovery studies in CRCs.

# Discussion

CRC is a heterogeneous disease, with subtypes that are driven by different types of genomic alterations: hypermutated tumors, tumors showing CIN, etc. Multiomics data analysis has the potential to clarify questions regarding disease subtypes. Taking advantage of this will require new methods which scale computationally as the amount of available data increases. In addition to stratifying patients into clinically relevant subgroups, it is necessary to find potential drug targets specific to each subtype. Most drug target discovery studies use cancer models such as cell lines, organoids, or xenografts, and it is thus necessary to match these cancer models to the appropriate subtype in each study, or if a cancer model is inappropriate for the study of any subtype, to be able to flag it as such.

We have developed an autoencoder-based method, called maui, that integrates data from multiomics experiments. We used it to infer latent factors that summarize molecular patterns, from data sets made up of RNA-seq, single nucleotide polymorphisms, and copy number variants (CNVs). The latent factors capture patterns which explain the variation across the different data modalities, capturing important aspects of cancer biology including different gene expression programs, mutational profiles, copy number profiles, and their interactions. We showed that these latent factors are predictive of previously described CRC subtypes (CMS). maui outperforms the methods MOFA and iCluster+, which we benchmarked against in the task of CMS label prediction and in predicting patient survival regardless of the CMS subtypes involved. From the standpoint of computational performance, maui can extract more latent factors from larger data sets at a fraction of the computational cost of both iCluster+ and MOFA, making it better suited for the analysis of increasingly larger data sets. It can leverage its computational efficiency to learn from larger data sets, containing more genes, to produce latent factors which are more predictive of patient survival; using 6,000 genes, maui produces more clinically relevant representations than using 1,300 genes. We also used these latent factors to produce a novel system of classification for CRC. In addition to

reproducing the CMS nearly perfectly, it revealed that one of the CMS subtypes, CMS2, is in fact two distinct tumor subtypes. These have different characteristics in terms of the underlying gene expression programs and distinct survival rates. This shows that the prognostic value is improved by an unbiased selection of more input genes, rather than restriction to known markers or driver mutations. Our results support the idea that passenger mutations and driver mutations could have an effect on cancer outcomes ([38]).

iCluster+, which we compared with maui in the first part of this study, is already strained at 1,300 input genes (runtime of 11 h), and in the future, with even more data types (e.g., methylation), we expect the input spaces to grow far beyond 6,000 that were used here. We still recommend that users of maui who are interested in using, for example, DNA methylation data perform some feature selection on the 450,000 or so CpG's, in a similar way to the feature selection we performed on gene expression, mutation, and copy number data. Hence, the computational efficiency of maui is not a mere academic exercise; at today's scale, this increase in computational efficiency is the difference between a model that can be fit at all and one which cannot. In addition to using more input features, the computational efficiency allows maui to learn more latent factors than we might believe to truly exist in the data. This is desirable in latent factor models, as fitting more latent factors increases the amount of ground truth factors which are recovered by a method. This effect tends to outweigh any harm that may come from overparameterizing the model ([15] *Preprint*). By example, ranking latent factors by their clinical relevance (as in Fig 3B), we have shown that we can fish out the ground truth latent factors from a potentially overparameterized model. So here too, the computational efficiency premium offered by maui over iCluster+ and MOFA comes with real-world benefits.

Individual latent factors have prognostic relevance. They can also be associated with genes. A pathway analysis of the latent factors that are most predictive of patient survival revealed an enrichment of known CRC-related pathways, such as Wnt signaling and other APC-mediated processes, NGF signaling, and PDGF signaling ([39]). Although the association between latent factors and individual genes is not as straightforward with maui as using matrix factorization methods, the pathways it reveal are informative. We also propose a way to use latent factors learned by maui to predict the overall fitness of specific cancer cell lines as models for CRC and specific subtypes of the disease. We hypothesized that cell lines which bore a higher resemblance to other cell lines than to CRC tumors would serve as poorer models. To test this, we included non-CRC cell lines in the sample. By testing whether a cell line is more similar to other (noncolon) cell lines or CRC tumors, we correctly identified 98% of non-CRC cell lines, which are very likely to be poor models for CRC. In addition, ~45% of CRC cell lines were predicted to be poor models for CRC, including COLO741 and COLO320, which are known to be inappropriate ([35], [36], [37]). These conclusions will need to be validated by further experiments. Although cell lines that were predicted to be less appropriate for the study or CRC may still be helpful in studying, for example, genetic interactions, their utility in studies of, for example, adaptive drug response may be limited. On the other hand, the SW480 and SW620 cell lines are predicted to be a good match for CRC, and they have shown similar drug response to clinical trials on *KRAS* mutant tumors ([40]).

By including the cell lines predicted to be appropriate in the clustering analysis, we also assigned them to specific CRC subtypes, a finding which is likely to have far-reaching potential for drug trials. One of the clusters (cluster 2, CMS1) consists mainly of hypermutated tumors with low CIN, and the cell lines we matched with that cluster using maui share their characteristics; such matches have been a standard method in finding disease-specific cell lines (41), and this shows that maui captures this characteristic. We hope that our approach to predicting the fitness of cancer cell lines as models for tumors can be verified and extended to other cancer models, such as organoids and xenografts, in future experiments. This could make maui indispensable for of drug discovery pipelines and speed up new therapeutics.

The CRC subtypes we used as a starting point for this study had previously been defined based solely on gene expression profiles. Our aim to refine these subtype definitions using multiomics data restricted us to a subset of the tumors used in the CMS study. We used only samples from the TCGA which had measurements for gene expression, mutations, and copy numbers (n = 519). The original CMS study used a larger cohort (n = 4,151). It is unclear whether our proposal of splitting the CMS2 subtype into two clusters would hold for a larger data set. Only when multiomics data are available, this question will be possible to address.

Although the autoencoder architecture of maui is able to perform inference in larger data at a fraction of the time required by matrix factorization methods such as MOFA and iCluster+, it is more challenging to provide a biological interpretation of the model it produces. Linking genes to latent factors is not as straightforward as in matrix factorization. The solution we propose is based on correlations between input genes and latent factor values, heuristically picking those that are most significant. Although we were able to show that such latent factors—gene relationships capture meaningful information about cancer biology and recapitulate known associations between the dysregulation of certain pathways and patient survival, this method is potentially less robust in establishing these associations and might require more user involvement in the analysis pipeline.

In this study, we have developed a deep learning–based multiomics integration method (maui) and shown that it can be used to define clinically relevant subtypes of CRC, as well as predict the fitness of cancer cell lines as models for the study of tumors. We found new associations between cell lines and particular CRC subtypes. The latent factors inferred by maui are also interpretable in a biological context and predictive of patient survival, which permits associations to be made between underlying oncogenic processes and patient survival. We benchmarked maui against two state-of-the-art methods for multiomics data integration and showed that not only is it more effective in defining clinically meaningful subtypes but also it does so with superior computational efficiency. An increase in speed of orders of magnitude will permit maui to be used in studies involving larger cohorts and more types of omics data, an experimental trend which will continue to increase in the future. maui's suitability as a general tool for multiomics integration should also make it useful outside of the context of cancer to explore issues related to in basic biology in studies using multiple genomic assays.

# Materials and Methods

## Data

We obtained data for tumors from the TCGA-COAD (n = 389) and TCGA-READ (n = 130) project designations of the Genomic Data Commons (https://portal.gdc.cancer.gov) using the *TCGAbiolinks* R package (42). We downloaded the CMS annotations for the TCGA tumors from the Colorectal Cancer Subtyping Consortium (CRCSC) (http://sagebionetworks.org/research-projects/colorectal-cancer-subtyping-consortium-crcsc/). Table 1 summarizes the subtype information. The gene expression data (mRNA) are HTSeq-FPKM. Mutations were downloaded as mutation annotation files, filtered to include nonsynonymous mutations only, and represented as a binary mutation matrix where $m_{ij} = 1$ if and only if gene $i$ carries a nonsynonymous mutation in sample $j$. Copy number alterations are GISTIC calls by gene, represented as a real-valued matrix where $c_{ij}$ is the GISTIC segment mean for the segment containing gene $i$ in sample $j$.

In addition, we obtained clinical metadata from the TCGA about the same patients. In addition to survival data, we used the age at diagnosis, sex, and tumor stage at diagnosis, as clinical covariates.

CCLE data were obtained from the CCLE portal (https://portals.broadinstitute.org/ccle) and are the same data types as the TCGA data, with the exception that transcriptome profiles are reads per kilo base per million mapped reads (RPKM)-normalized and not fragments per kilo base per million mapped reads (FPKM). We considered 54 cancer cell lines originating from the colon.

We considered only tumors (from TCGA) and cancer cell lines (from CCLE) which have "complete data", that is, available measurements in all three assays: gene expression, SNVs, and CNVs.

We used gene-wise median absolute deviation statistic, computed directly on the raw data described above, to select the most informative genes. For the comparisons with MOFA and iCluster+, we used the 1,000 genes with the highest median absolute deviation for gene expression, 200 for mutations, and 100 for copy number alterations, for a total of 1,300 input features. We selected the features so strongly to make a comparison against iCluster+ viable, and with a larger feature space, the runtime would become untenable (Table 2).

For the final clustering analysis, we used a larger feature space, with 5,000 gene expression values, 500 mutations, and 500 CNVs for a total of 6,000 features, taking advantage of maui's neural network architecture which allows for larger feature spaces to undergo feature selection as part of the training.

We fit the autoencoder using all TCGA samples, both with and without a CMS label (n = 519, Table 1) as well as colon-derived cancer cell lines (n = 54), for a total training set size of 573. For the analysis that depends on a CMS label being available, the input features were the latent factors and the samples only those TCGA samples with a well-defined CMS label (n = 419, see Table 1).

All input features were scaled and centered before feeding to the neural network, using batch normalization. Before this scaling, mutation data were binary, CNV data GISTIC calls, and gene expression counts were RPKM/FPKM values which were scaled and centered. TCGA and CCLE gene expression matrices were first scaled and centered individually and, then, concatenated and scaled jointly to filter out the "batch effect" of CCLE versus TCGA data

enable mapping of tumors and cell lines to the same space. This means that for a trained maui model, when new, unseen samples are to be mapped onto the latent factor space, they must first be normalized in this way to fit the distribution of the training data.

## Network-smoothing of multiomics data

We applied *netSmooth* (27) to the binary mutation matrix before feeding it into the neural network of maui. The method uses the PPI to smooth noisy molecular assays in effect, incorporating prior data from countless previous experiments, to improve the signal-to-noise-ratio. The intuition behind the method is that genes seldom act alone, and genes in close neighborhoods in the PPI are expected to behave similarly. For instance, interacting proteins tend to be coexpressed (43), and somatic mutations or amplifications/deletions in interacting may lead to similar dysfunctions. We used $\alpha = 0.7$, a rule-of-thumb for network smoothing mutations (26).

The algorithm is a simple *Random Walks with Restarts* diffusion process on the PPI, described by the iterative process

$$F_{t+1} = \alpha A F_t + (1 - \alpha) F_0,$$

where $F$ is a data matrix (gene expression, mutations, etc.), $A$ is the degree-normalized adjacency matrix of the PPI, and $(1 - \alpha)$ is the restart rate. The process is guaranteed to converge and has a closed-form solution

$$F_\infty = (I - \alpha A)^{-1} F_0.$$

To pick the optimal $\alpha$ value, we performed a grid search over a range between 0 and 1, and picked the lowest $\alpha$ value within 1 SD of the highest score on the Harrell's c-index benchmark (Fig 1E–G).

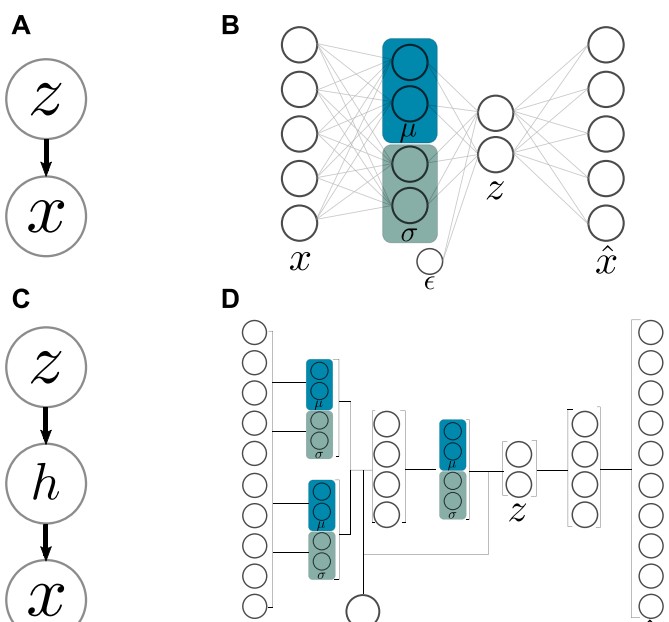

**Figure 7. Variational autoencoder. (A)** Plate model of latent factor model; $x \sim p(z|x)$. **(B)** Visual representation of the VAE. **(C)** Plate model of the stacked latent factor model. **(D)** Visual representation of the stacked variational autoencoder.

## Latent factor model for multiomics data

Starting from different data matrices $x_i$ from different modalities, we call the full multiomics data set $x = [x_1, x_2, ..., x_m]$.

We define a generative model $x \sim p(x|z)$. Graphically, our model looks like Fig 7A, a Bayesian latent variable model where the variation in the data $x$ are explained by the variation in a smaller set of latent factors, $z$. To infer the latent variables $z \sim p(z|x)$, as $p(z|x)$ is generally intractable, we proceed with a variational Bayes framework, that is, approximating $p(z|x) \approx q_\theta(z|x)$, where $q_\theta(z|x)$ is a simple class of distribution, and minimizing the Kullback–Leibler (KL) divergence $D_{KL}(q_\theta(z|x) \| p(z|x))$. This is equivalent to maximizing the evidence lower bound (44):

$$ELBO = E_q\big[log\big(p_\phi(x|z)\big)\big] - D_{KL}\big(q_\theta(z|x) \| p_\phi(z)\big).$$

In the equation above, $log\big(p_\phi(x|z)\big)$ is the log-likelihood of the reconstruction, so that its expectation under the encoder, the first term $E_q\big[log\big(p_\phi(x|z)\big)\big]$ is the cross-entropy loss of the reconstruction. Thus, the second term (KL divergence) can be seen as a regularizer on the latent factors.

We follow (16 *Preprint*) and reparameterize $z_i^l$ as

$$z_i^l = \mu_i + \sigma_i \epsilon_l$$

where

$$\epsilon_l \sim N(0, I),$$

which allows us to construct the autoencoder shown in Fig 7B.

The first half of the autoencoder, leading from $x$ to $z$ (the "encoder"), is a neural network which will be trained to compute $q_\theta(z \lor x)$, that is, $\theta$ denotes the weights of the encoder network. The second half, the "decoder" network, is a neural network which will be trained to compute $p_\phi(z \lor x)$, so $\phi$ denotes the weights of the decoder network. Thanks to the reparametrization of $z$, the path from $x$ to $\hat{x}$ is differentiable, via backpropagation, in $\theta$ and $\phi$, and thus, we can use gradient descent to optimize a loss function that is differentiable in $\theta$ and $\phi$.

Setting the loss function of the neural network to the negative evidence lower bound

$$l = -E_q\big[log\big(p_\phi(x|z)\big)\big] + D_{KL}\big(q_\theta(z|x) \| p_\phi(z)\big),$$

we see that the first term is equivalent to the cross-entropy reconstruction loss, and the second term, the KL-divergence between $q_\theta(z|x)$ and the prior $p_\phi(z)$ can be seen as a regularization term, will push the $z$'s to their prior distribution.

## Stacking autoencoders

The VAE described above is for a one-layer Bayesian framework, that is, Fig 7A. But autoencoders may be stacked (45) to produce deeper neural network architectures. Deep architectures have more than one layer of nonlinearities and can, thus, more compactly capture highly nonlinear functions. We introduce a hidden layer to our Bayesian latent variable model (Fig 7C).

Using the reparametrization trick as above and specifying the full loss function, inference in the generative model (Fig 7C can be performed by backpropagation in the stacked VAE model (Fig 7D).

## Model regularization

Deep neural networks have many parameters, making them very flexible. This flexibility, however, comes at a cost—deep models are prone to overfitting: the generation of models which explain the training data well, but generalize poorly to new data. In addition, deep nets are prone to producing complex relationships between many variables. In the case of a latent variable model, that means latent factors that change with the variation of any of a large number of input features, a property which makes the task of interpreting the biological meaning of those latent factors difficult. In technical terms, we wish to enforce sparsity in $q_\theta(z|x)$ so that each latent factor will depend on fewer of the inputs.

To address the first issue of potential overfitting, we used batch normalization (46). When fitting the model, we segmented the data into minibatches, at each iteration computing derivatives and making updates to the model based on that sample. Using batch normalization, each feature is scaled and centered in each minibatch. We fed all of the training examples to the model fitting procedure until the entire training set is exhausted, and then, we segmented it into new minibatches and repeated the process, for a specified number of epochs. This way, each time a training sample is passed to the model, it will be slightly different, which is roughly equivalent to adding noise, which has been shown to work as a regularizer in denoising autoencoders (47) and prevent overfitting. Furthermore, batch normalization addresses another issue—that of internal covariate shift. Internal covariate shift happens when the distributions of activations of internal nodes in the neural network change while training. Reducing internal covariate shift enables us to pick higher learning rates and, thus, speeds up inference considerably.

The second mode of regularization, encouraging representations, where latent factors depend only on a few input features, is partially achieved by the KL term in the loss function, as that penalizes distributions of $z$'s which are far from the Gaussian prior. Disentangled representations, where latent factors depend on complementary input feature sets, can support this kind of sparsity. This holds when latent factor representations are non-negative, which we achieve by passing them through a rectified linear unit (ReLU). When each non-negative latent factor depends on a different set of inputs, the relationships will be sparser.

We, therefore, add a multiplier to the loss function similar to $\beta$-VAE (18) and, allowing us to weigh the relative importance of the terms:

$$l = -E_q\big[log\big(p_\phi(x|z)\big)\big] + \beta D_{KL}\big(q_\theta(z|x) \parallel p_\phi(z)\big)$$

To ensure the network finds a good representation before it starts regularizing, we used the "warm-up" method proposed by (17 Preprint), where $\beta$ is initially 0 and is gradually increased by $\beta = \beta + \kappa$ until its value reaches 1.

## Model selection

The stacked VAE presented above is a class of models which are parameterized by the number of hidden units (the dimensionality of $h$), $N_{hidden}$, and the number of latent factors, $N_{latent}$.

This presents an opportunity for selecting the best model by spanning a grid over the two parameters and computing some scores. We searched the space spanned by ($N_{hidden}$, $N_{latent}$) and computed a compound benchmark score at each point. The compound benchmark score is the average of the scores of the AUC in the supervised CMS prediction task, the AMI in the unsupervised CMS subtype prediction task, the $-log_{10}p$ of the multivariate log-rank test for differential survival statistics, and the c-index (25) from the Cox proportional hazards model. maui is largely insensitive to the choice of ($N_{hidden}$, $N_{latent}$), for $N_latent$ > 30 (Fig S10).

MOFA was run using the default parameters. It uses heuristics to pick the number of latent variables, starting with a large number and pruning away ones with an explained variance ratio of beneath a threshold of 2%. The resulting model had 20 components. To see if MOFA's heuristic picks a sensible model, we also ran it with fixed numbers of latent factors over a range from 10 to 30 and computed the composite benchmark, lower than for maui because of the higher runtime.

For iCluster+, there are two free parameters: the regularization parameter $\lambda$ and the number of latent factors. We ran a grid search over the regularization parameter and number of latent variables space, similar to the way maui was tuned, but with a lower number of maximum latent factors because of iCluster+'s prohibitive runtime for larger numbers. For each parameter configuration, we computed the compound benchmark. maui consistently outperforms both MOFA and iCluster+ for most parameter sets (Fig S11).

For the final analyses shown in the results section, to avoid leakage of benchmarks into the unsupervised learning algorithms, we ran maui with parameters corresponding the mean of the distribution of compound benchmarks ($N_{hidden}$ = 1,100 and $N_{latent}$ = 100); the same reasoning for iCluster+ resulted in five latent factors. We allowed MOFA to use its own heuristic, discarding latent factors with variance explained below 2%, yielding a 20-component model. We used the MOFA default threshold when picking the number of components to keep in the PCA comparison, which yielded five components.

## Model implementation

We implemented the model using Keras (v2.1.5) using a TensorFlow (v1.6.0) backend. We used rectified linear units for all activations except for the last layer which is sigmoids, for all features. We trained our network for 600 epochs using minibatches of size 100 and $\kappa$ = 0.01. We used the Adam optimizer (48).

## Predicting CMS from latent factors using SVM

To quantify the correspondence between latent factors learned by using different methods and the CMS label, we used SVMs (49), a supervised learning algorithm. There were two levels of CV; first, we split the whole data set into 10 folds, reserving at each time 10% of the data set as a test set. At each round, we trained the latent factor models (maui, iCluster+, and MOFA) and the SVM predictor on 90% of the data and used that model to assign a CMS subtype to the

remaining 10%. Within each CV fold, we used regularized SVM to predict the CMS from the latent factors. The regularization parameter of the SVM was picked using another 10-fold CV, splitting the "outer" 90% training data into "inner" ninety 10 splits. ROC curves were computed for each class by modeling a binary outcome for each CMS label (one-versus-all). Mean ROC curves were computed by averaging the ROC of all CMS labels at each point.

### Unsupervised prediction of CMS from latent factors using k-means clustering

We benchmarked maui against MOFA and iCluster+ in the power of latent factors to predict CMS labels in an unsupervised fashion to present a fair comparison between maui (70 latent factors) and MOFA (20 latent factors) and iCluster+ (10 latent factors). We used k-means clustering, as clustering based on distance metrics suffers from the "curse of dimensionality," and does not, in general, benefit from a larger number of input dimensions (unlike supervised learning methods). To assess the ability of k-means clusters to capture the CMS labels, we ran k-means with 1,000 starts, picking the best (lowest variance) solution for each run. In addition, we applied the algorithm with $K$'s in the range of 2–9. For the cluster assignments for each $K$, we computed AMI of the clustering with the CMS labels. The AMI is an information-theoretic measure of the concordance between two labelings (clusterings and CMS), which accounts for chance. Higher values indicate closer relationships between labelings.

### A novel subtyping scheme for CRC with cell line associations to subtypes

The subtyping scheme presented in the results section is based on k-means clustering using maui latent factors learned from multiomics data. We did this using a maui model trained on 6,000 input features (5,000 gene expression, 500 mutation, and 500 CNV), as it is more predictive of patient survival than the one using 1,300 features (Fig 1F) and produces largely the same cluster assignments as the 1,300 gene model presented above (Fig S12).

### Association of latent factors with genomic features

The stacked variational autoencoder model described above computes latent factors $z = f(x)$, where $f(x)$ is a nonlinear function which may not necessarily be well approximated by a linear $z \approx Wx$, as in models such as MOFA or iCluster+. The architecture and depth of the neural network also makes it nontrivial to associate the input genomic features (gene expression, mutations, etc.) with the different latent factors. However, to make biological sense of the latent factors, it is necessary to make that association. To do that, we computed Spearman's $\rho$ for each latent factor with each input feature and called a latent factor associated with an input feature if $P < 0.001$.

### Gene set enrichment

To identify genes associated with the different clusters, we performed a differential expression analysis using $t$ tests and Benjamini–Hochberg correction for multiple hypothesis testing. Genes with adjusted $P$ value below 0.05 were called differentially expressed. To find out if the genes associated with latent factors (Fig 3) or with clusters (Fig 2) belong to known pathways, we used *Enrichr* (50, 51) via the python package *gseapy* (version 0.9.4, available from PyPI https://pypi.org/project/gseapy). We used pathways (gene sets) defined by the Kyoto Encyclopedia of Genes and Genomes (52, 53, 54).

### Survival analysis

We relied on overall survival data from the TCGA annotations for all survival analyses.

To assess the prognostic value of latent factors inferred by our deep learning approach, we fit a Cox proportional hazards model (29),

$$ln \frac{h(t)}{h_o(t)} = \sum_i \beta_i x_i,$$

where the left hand side is the logarithm of the hazard ratio and $x$'s are covariates. We assessed the predictive value of each latent factor separately, while controlling for the patient's age, gender, and tumor stage at diagnosis. We computed confidence intervals for the coefficient $\beta$ associated with the latent factor and picked the latent factors with false discovery rate-correction and $\alpha = 0.95$.

To compare the prognostic value of different models, we computed the c-index (55, 56, 57) and used 5-fold CV (49).

The log-rank statistics reported in Figs 1D and S3 are multivariate log-rank test, under the null hypothesis that all groups have the same survival function, with an alternative hypothesis that at least one group has a different survival function.

All survival analysis was performed using the python package *lifelines* (https://lifelines.readthedocs.io/en/latest/).

### Comparing models' survival-predictive value

To compare maui to MOFA and iCluster+ (as well as to a gene expression only–based maui model), we used Harrell's C (25) in a Cox proportional hazards (29) regression model. The c-index was computed for Cox models based only on clinically relevant factors, which we selected using individual, unregularized Cox models, one per factor, while controlling for patient age, sex, and tumor stage. In those individual factor models, we used Efron's method to compute confidence intervals and only to keep the latent factors with statistically significant (adjusted $P$-value 0.05) nonzero coefficients in the individual Cox models. Having selected clinically relevant latent factors from each model (maui, MOFA, iCluster+, maui-expression, and maui-netsmooth), we fitted a full Cox regression using those 2and ran a cross-validated out-of-sample c-index calculation using regularized Cox PH regression, searching for the optimal result among the regularizers 1, 10, 100, 1000, 10000. The results reported in Fig 1F are the best-regularized model for each of the methods.

### Quality assessment of CRC cell lines for modeling tumors

To assess the fitness of different cancer cell lines as models for tumors, we computed the pairwise Euclidean distance between

each of the samples (TCGA and CCLE), in the space of the latent factors derived from maui. Then, we computed, for each cell line, the proportion of its five nearest neighbors which are also cell lines, the working hypothesis being that cell lines that form "cell line clusters" are more cell-line like than tumor like, and likely less fit as models for tumors. We repeated the exercise considering other numbers of nearest neighbors from 1 to 20, at each $K$ computing the true positive rate (recall), that is, $\frac{\text{No. of noncolon cell lines predicted to be poor models}}{\text{No. of noncolon cell lines}}$, showing that the recall is near perfect for a wide range of $K$'s.

## Software

maui is available as a general purpose python package for the study of multiomics data. For more information, visit https://github.com/BIMSBbioinfo/maui.

# Supplementary Information

# Acknowledgements

The work shown here is based on data generated by The Cancer Genome Atlas Research Network (http://cancergenome.nih.gov/) and the Cancer Cell Line Encyclopedia (CCLE) (https://portals.broadinstitute.org/ccle). We would like to thank Claudia Baldus, Gaetano Gargiulo, Vedran Franke, Bora Uyar, Wolfgang Kopp, and Ella Bahry for valuable comments on the manuscript.

## Author Contributions

J Ronen: software, formal analysis, visualization, methodology, and writing—original draft, review, and editing.
S Hayat: data curation, formal analysis, and writing—review and editing.
A Akalin: conceptualization, resources, data curation, supervision, funding acquisition, methodology, and writing—original draft, review, and editing.

## Conflict of Interest Statement

The authors declare that they have no conflict of interest.

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
