## [Reviewer comments · Life Science Alliance]

Life Science Alliance

Evaluation of colorectal cancer subtypes and cell lines using deep learning

Jonathan Ronen, Sikander Hayat, and Altuna Akalin

DOI: <https://doi.org/10.26508/lsa.201900517>

Corresponding author(s): Altuna Akalin, Berlin Institute for Medical Systems Biology

Review Timeline:

Submission Date:	2019-08-07
Editorial Decision:	2019-08-07
Revision Received:	2019-09-20
Editorial Decision:	2019-10-21
Revision Received:	2019-11-04
Accepted:	2019-11-05

Scientific Editor: Andrea Leibfried

Transaction Report:

Please note that the manuscript was previously reviewed at another journal and the reports were taken into account in the decision-making process at Life Science Alliance. Since the original reviews are not subject to Life Science Alliance's transparent review process policy, the reports and author response cannot be published.

August 7, 2019

Re: Life Science Alliance manuscript #LSA-2019-00517-T

Dr Altuna Akalin
Berlin Institute for Medical Systems Biology
Max Delbrueck Center for Molecular Medicine
Robert Roessle str 10
Berlin 13125
Germany

Dear Dr. Akalin,

Thank you for submitting your manuscript entitled "Evaluation of colorectal cancer subtypes and cell lines using deep learning" to Life Science Alliance. The manuscript was assessed by expert reviewers at another journal before and the reviewers transferred those reports to us with your permission.

The reviewers appreciated the method, but were concerned that the approach was not fully unsupervised. They also thought that further validation of the framework would be needed. We would like to offer publication of this work in Life Science Alliance based on the reports already at hand and pending revision. Please provide a point-by-point response and accordingly changes to the manuscript text and discussion. The following specific concerns of the reviewers should also get addressed: Rev#1, point on maui performance (third paragraph), point on inconsistency (forth paragraph); Rev#2, point on cross-validation.

The typical timeframe for revisions is three months.

Thank you for this interesting contribution to Life Science Alliance. We are looking forward to receiving your revised manuscript.

Sincerely,

Andrea Leibfried, PhD
Executive Editor
Life Science Alliance
Meyerhofstr. 1
69117 Heidelberg, Germany

t +49 6221 8891 502
e a.leibfried@life-science-alliance.org
www.life-science-alliance.org

B. MANUSCRIPT ORGANIZATION AND FORMATTING:

October 21, 2019

RE: Life Science Alliance Manuscript #LSA-2019-00517-TR

Dr. Altuna Akalin
Berlin Institute for Medical Systems Biology
Max Delbrueck Center for Molecular Medicine
Robert Roessle str 10
Berlin 13125
Germany

Dear Dr. Akalin,

Thank you for submitting your revised manuscript entitled "Evaluation of colorectal cancer subtypes and cell lines using deep learning". Please excuse the delay in getting back to you. I had asked the original reviewer #2 to take a look at your revised manuscript for a technical re-review of the revision points I had previously outlined to you, and I had to give the reviewer more time due to her/his very busy schedule.

As you will see, the reviewer appreciates your response and the introduced changes and we'd be thus happy to publish your paper in Life Science Alliance pending final revisions necessary to meet our formatting guidelines:

- please upload all figures, also the supplementary figures, as individual files and without legends; all figure legends and tables should remain in the main manuscript text file
- please provide the manuscript text in docx format
- please mention all depicted panels in the figure legends (eg, see Fig S3, S4, S5)
- please list 10 authors et al in the reference list (you can use EMBOreports style in case you are using a reference manager)

A. FINAL FILES:

- An editable version of the final text (.DOC or .DOCX) is needed for copyediting (no PDFs).

B. MANUSCRIPT ORGANIZATION AND FORMATTING:

Sincerely,

Andrea Leibfried, PhD
Executive Editor
Life Science Alliance
Meyerhofstr. 1

69117 Heidelberg, Germany
t +49 6221 8891 502
e a.leibfried@life-science-alliance.org
www.life-science-alliance.org

Reviewer #2 (Comments to the Authors (Required)):

The revised manuscript addresses my previous concerns adequately, and I recommend acceptance.

November 5, 2019

RE: Life Science Alliance Manuscript #LSA-2019-00517-TRR

Dr. Altuna Akalin
Berlin Institute for Medical Systems Biology
Max Delbrueck Center for Molecular Medicine
Robert Roessle str 10
Berlin 13125
Germany

Dear Dr. Akalin,

Thank you for submitting your Methods entitled "Evaluation of colorectal cancer subtypes and cell lines using deep learning". It is a pleasure to let you know that your manuscript is now accepted for publication in Life Science Alliance. Congratulations on this interesting work.

DISTRIBUTION OF MATERIALS:

Again, congratulations on a very nice paper. I hope you found the review process to be constructive and are pleased with how the manuscript was handled editorially. We look forward to future exciting submissions from your lab.

Sincerely,
